# Stratum Ventilation: Enabling Simultaneous Energy Conservation and Air Purification in Subway Cars

**DOI:** 10.3390/ijerph192114521

**Published:** 2022-11-05

**Authors:** Yanhui Mao, Shengxu Wang, Jianzhou Liang, Saiqin Mao, Yukun Han, Shengquan Zhang

**Affiliations:** School of Architecture and Transportation Engineering, Ningbo University of Technology, Ningbo 315211, China

**Keywords:** subway car ventilation system, stratum ventilation, contaminant removal, dilution purification, energy conservation, coronavirus disease pandemic, conventional ventilation, computational fluid dynamics, air conditioning

## Abstract

The supply of fresh air for underground rail transit systems is not as simple as opening windows, which is a conventional ventilation (CV) measure adopted in aboveground vehicles. This study aims to improve contaminant dilution and air purification in subway car ventilation systems and the safety of rail transit post-coronavirus disease pandemic era. We designed an air conditioning (AC) terminal system combined with stratum ventilation (SV) to enable energy consumption reduction for subway cars. We experimentally tested the effectiveness of a turbulence model to investigate ventilation in subway cars. Further, we compared the velocity fields of CV and SV in subway cars to understand the differences in their airflow organizations and contaminant removal efficiencies, along with the energy savings of four ventilation scenarios, based on the calculations carried out using computational fluid dynamics. At a ventilation flow rate of 7200 m^3^/h, the CO_2_ concentration and temperature in the breathing areas of seated passengers were better in the SV than in the CV at a rate of 8500 m^3^/h. Additionally, the energy-saving rate of SV with AC cooling was 14.05%. The study provides new ideas for reducing the energy consumption of rail transit and broadens indoor application scenarios of SV technology.

## 1. Introduction

Urban rail transit is a vital means of alleviating urban traffic congestion and can optimize the layout of urban functions. Statistics from the International Association of Public Transport indicate that the total mileage of rail transit is constantly increasing globally. During 2018–2020, countries such as China, India, Australia, and Indonesia, added 3300 km of operating mileage [1]; therefore, reducing energy consumption for rail transit can be a crucial climate change initiative for the transportation sector [2,3]. Due to climate differences in various regions, the energy consumption of rail transit air-conditioning (AC) systems accounts for 19.4–30% of the operational energy consumption [4,5]. Ventilation and AC not only affect passenger comfort in subway cars but also impact the transmission of respiratory contaminants. As rail transit systems predominantly operate underground, the supply of fresh air is not as simple as opening windows, which is a common ventilation measure adopted in aboveground vehicles. The effects of heavy metal pollution [6], particulate matter [7,8], and other contaminants on the air quality of passenger cars are garnering interest among researchers. Notably, the coronavirus disease (COVID-19) pandemic has impacted the public sense of safety regarding subway travel [9,10]. During 2020, passenger volumes for rail transit in Asia-Pacific and North American regions decreased by 32% and 63%, respectively, compared with those in 2019, and these numbers have yet to recover to pre-pandemic levels [1]. The reductions in passenger traffic have increased the operating costs of public transport systems. For example, the operating costs of 20 metro lines in the Beijing urban rail transit system increased by 67.1% in 2020 compared with 2019 [11]. Increasing the supply of fresh air [12] and installing air purification [13,14,15] and disinfection [16,17] devices are common and effective ways to reduce contaminant concentrations and improve air quality in subway cars. Previous studies have proposed the use of car-interior temperature detection sensors to monitor the heat load, achieve dynamic heat exchange, and promote energy conservation [18]. In most systems, porous copper plates are set in air channels to form a ventilation and sterilization device that can purify air [19]. Additionally, a ventilation system that directs air (containing airborne infectious bacteria, viruses, and other pathogens) in chambers fitted with ultraviolet (UV) lamps placed at the top of cars to ensure radiation sterilization has been designed [20]. Although these solutions help purify air and remove contaminants, they also increase the energy consumption of subway cars’ AC systems. Notably, improving the air quality of subway cars and reducing the energy consumption of their AC systems are contradictory objectives, and the ideal method to achieve both simultaneously remains to be devised.

The distribution of respiratory pollutants in subway cars has significantly different characteristics from that in indoor spaces. There are both standing and sitting passengers in subway cars; therefore, the contaminants exhaled by passengers at different heights are stratified within the cars and entrained by the airflow of the ventilation system. Consequently, the exhalations from standing passengers flow to the areas where seated passengers, including children, are breathing, resulting in cross-contamination. Improving the airflow organization of AC systems in subway cars is a recent concept that aims to improve air quality and reduce energy consumption. A previous study proposed a displacement ventilation model for subway cars [21], in which air supply vents are located under the seats, and exhaust vents are located on the ceiling of the carriage. Air would then flow from the bottom to the top due to thermal buoyancy, blowing the contaminants upward and discharging them from the upper area of the car. However, this resulted in the contaminants being attached to the ground to float upward again [22]. Stratum ventilation (SV) is a new technology that requires vents to be placed in areas where people are active to effectively dilute pollutants [23,24] while reducing the supply of fresh air to unoccupied areas to save energy [25,26]. Current research on the application of SV has focused on large spaces such as offices and has yielded some design parameters and operational guidelines [27,28,29]. However, in subway cars, passengers exhale pollutants at different heights in a confined space, and there is a cross-transmission risk of pollutants along the various height levels. Therefore, the principle of pollutant spread is different from that of a single height in the office space. The existing studies on the energy conservation and clean performance of SV are not suitable for subway cars, and the current study addresses this gap.

## 2. Materials and Methods

### 2.1. Technical Solution

In conventional ventilation (CV) systems installed in subway cars (Figure 1a), fresh air enters from the carriage ceiling and circulates inside the car. A portion of the circulated and contaminated air is discharged through the intersection of the ceiling and the body and the remainder from the underside of the seats. In this study, we designed an SV system with mid-height air vents [30], suitable for subway cars, as shown in Figure 1b. Compared to the CV system, the SV system increases air supply in the middle of the subway body to provide fresh air to seated passengers so that two ventilation-flow areas are formed in the carriage. In the upper airflow area, air enters from the ceiling of the carriage and discharges through the intersection of the ceiling and the body. The lower airflow enters from the central air inlet and is discharged from the underside of the seats. This airflow division is maintained to avoid the intersection of the upper and lower airflows. The research shows that the SV system in subway cars is effective in reducing the probability of respiratory virus transmission [31].

### 2.2. Study Method

Computational fluid dynamics (CFD) was used to study the effects of SV on contaminants dilution, air purification, and energy conservation in subway cars. Jiang et al. [32] applied the K-epsilon (k-ɛ) turbulence model to analyze the relationship between the probability of doctor–patient transmission and the ventilation flow rate in Xiaotangshan Hospital during the SARS pandemic. Shen et al. [33] used the k-ɛ model to analyze various combinations of personalized and background ventilation systems on contaminants and occupants’ comfort in offices. The large eddy simulation (LES) [34] and the shear stress transport (SST) k-omega (k-ω) [35] models have been used in numeric simulation studies to calculate the diffusion of indoor contaminants, each with its own computational advantages. In this study, similarities between physical parameters and propagation characteristics of the exhaled contaminants in subway cars and airflow in hospitals [32] and offices [33] were considered. The flow of contaminants exhaled by passengers in cars with ventilation systems was compared using the k-ɛ model to derive reliable experimental findings. The finite volume method was used to discretize the fluid equation, whereas the velocity and pressure were coupled based on the semi-implicit method for pressure-linked equations. A species transport model was also used to simulate the interactions between the components and diffusion trajectories of the contaminants under unsteady conditions. The computational domain is discretized into approximately 8.5 million cells using unstructured grid, and local refinements are performed at the nostrils, air supply inlet, and exhaust outlet. The CFD method simulated the pollutant distribution of two people breathing continuously for 180 s in the subway car. If the duration is too short, the cross-contamination characteristics of the upper and lower pollutants cannot be fully displayed; if the duration is too long, the pollutants’ diffusion process is difficult to capture.

As subway heating mainly consists of heated seats, AC is not the primary heat source. Therefore, summer AC cooling was selected as the ideal study condition; the selected subway car model was one-quarter of the length of a Type-B train, with a whole-car ventilation flow rate of 8500 m^3^/h, which was in line with the design parameters of an indoor per-person air supply of 30 m^3^/h (Type-B trains generally carry 240–330 passengers), as established in the Design Code for Heating Ventilation and Air Conditioning in Civil Buildings GB50736-2012. In order to compare the effects of energy conservation and pollutants dilution between SV and CV subway cars, the air supply flow rate of the SV subway car was designed for three working scenarios of 8500, 7850, and 7200 m^3^/h, based on 240 people per carriage. The ceiling and middle air supply flow rate for the SV subway car was 1:1, based on an equal number of seated and standing passengers. Table 1 lists the boundary and initial conditions of the SV subway car.

The number and location of passengers in the compartment have an important influence on the propagation of pollutants. When there are too many people, bodies will block the airflow and cause many possible flow directions, which is not conducive to capturing the interaction between ventilation and exhaled pollutants. Many researchers [36,37] have developed two-person models in enclosed spaces to better demonstrate the trajectory of airflow movement, and this study draws on this approach. In addition, droplets produced by coughing [38,39] and exhaled CO_2_ [40,41] are both key research targets for respiratory pollutants. As a new technology, SV in subway cars requires measuring and verifying the phenomenon of adulteration layered propagation in the vehicles. Since CO_2_ is more accessible to measure than droplets, CO_2_ was used to represent exhaled pollutants.

### 2.3. Model Validation

Spatial concentration and distribution of CO_2_ for one standing and one seated passenger’s breathing in the subway car in conditions of active AC cooling were tested and numerically simulated to verify the applicability of the turbulence model. The exhaled CO_2_ concentrations in the car were monitored at various heights using six Narrow-Band Internet of Things (NB-IoT) sensors. Figure 2 portrays a field photograph of the experimental test, a geometric model of the cabin ventilation system, and six instrument measuring points. The experimental test results were compared with the simulated values using CFD. Figure 3 portrays the values averaged from three consecutive tests. The polynomial fit curve was consistent with the overall trend of the curve calculated using CFD. Notably, the CO_2_ peaks were observed at heights of 800 and 1550 mm, which roughly corresponded with the breathing areas of the seated and standing passengers, respectively, indicating a bimodal distribution of exhaled contaminants in the car. The concentrations of CO_2_ outside the breathing zones were lower than that inside the breathing zones due to airflow diffusion. The peak error between the experimental and CFD calculations was around 12%, indicating the effectiveness of the CFD method in analyzing exhaled contaminant flow in subway cars.

## 3. Results

### 3.1. Dilution Purification Performance

Contaminant removal effectiveness (*CRE*) is a common index used to measure the ability of a ventilation method to eliminate contaminants [42]. As shown in Equation (1), when the contaminant concentration, *C*_e_, at the air outlet equals that at point P (*C*_p_) of the indoor contaminant source, then *C*_e_ = *C*_p_ and *CRE* = 1, which is equivalent to all the contaminants released by the source per unit time that passes through the air outlet.
(1)$$CRE=\frac{C_{e}-C_{s}}{C_{p}-C_{s}}$$
where *CRE* is the contaminant removal effectiveness at point P in the ventilated space, and *C*_s_, *C*_e_, and *C*_p_ are the contaminant concentrations at the air inlet, air outlet, and point P, respectively.

Because the air inlet of the AC system did not contain CO_2_, the passenger breathing area was considered the emission point of the contaminant source. Thus, Equation (1) can be re-written as Equation (2), as follows
(2)$$CRE=\frac{C_{e}}{C_{p}}$$

We compared the differences in the air distribution between the CV and SV cars and assessed their respective airflow velocity fields (Figure 4). The ceiling air supply of the CV car circulated from the top to the floor, resulting in the exhaled contaminants being entrained and transmitted from the upper air layer to the lower ones. In the SV car, fitted with a middle air outlet, fresh air was supplied to the seated passengers, with each inlet providing half of the total air supply flow rate. The middle inlet formed an air curtain above the seated passengers, blocking the downward transmission of the contaminants exhaled by the standing passengers, thereby dividing the airflow in the car into upper and lower airflow areas. This prevented the crossover of contaminants exhaled in the upper and lower layers.

Figure 5 compares the dilution effects of CV and SV cars, with CO_2_ distribution being 8500 m^3^/h in the CV car and 8500, 7850, and 7200 m^3^/h in the SV car. We measured the area-weighted average concentrations of CO_2_ in a 20-cm-diameter circular area in front of the faces of the standing and seated passengers (Figure 5a; P_o,t_ and P_o,s_) and the upper exhaust and under-seat outlet (P_e,t_ and P_e,b_), to quantify the *CRE* of the two ventilation methods.

According to the nephogram of CO_2_ concentrations shown in Figure 5, the flow trail of CO_2_ exhaled by the seated passengers in the CV car portrayed a tendency to float upwards and gradually toward the breathing area of the standing passengers. The CO_2_ exhaled by the seated passengers in the SV car was guided downward by the middle air inflow and discharged under the seat, shortening the retention time of the contaminants in the car. Based on the concentrations (Figure 5b–d) and using SV, the speed of the ceiling air supply decreased with the ventilation flow rate, and the concentration of CO_2_ in front of the standing passengers’ faces remained almost unchanged. However, the spatial diffusion area of the CO_2_ concentration increased, which was due to the weakening CO_2_ entrainment caused by a slow air supply to the ceiling. When the ventilation rate was 7200 m^3^/h, the middle air inlet of the SV system continued to improve the air quality in the seated breathing area. The CO_2_ concentration in the seated breathing area in the SV car was 0.001%, which was lower than that in the same area in the CV car (0.0035%), reflecting the advantage of a targeted air supply using SV.

Based on the normal state of the passengers in the subway car, two mannequins were built, with one standing and one sitting person in the carriage (see Figure 2). The bar chart in Figure 6 portrays CO_2_ concentrations corresponding to the four monitoring points for the four calculation scenarios depicted in Figure 5. In this example, the CO_2_ concentration in the breathing area of the standing passenger was the lowest in the CV car because the entire air supply was supplied from the inlet at the top of the car, which provided a better dilution effect of the contaminants in the upper part of the car. In the SV car, as the top and middle inlets each supplied 50% of the airflow rate, the CO_2_ concentration in the breathing area of the standing passengers was slightly higher than that in the CV car. The seated passengers were supplied with fresh air in the SV car, so the CO_2_ concentration in the breathing area of the seated passengers was better in the SV car than in the CV car. Furthermore, as the CO_2_ exhaled from seated passengers was entrained by the middle air supply and discharged at the air outlet under the seat, the contaminants exhaled by the seated passengers did not float upwards, which occurred in the CV car (Figure 5a). Thus, the contaminant concentration at the lower air outlet was higher in the SV car than in the CV car, with the CO_2_ concentrations at the air outlets being essentially equal in the three ventilation rate scenarios; notably, all had relatively high *CRE*s.

We used Equation (2) to calculate the *CRE* in the CV and SV cars. As the seated passengers were supplied with fresh air in the middle of the SV car, the CO_2_ concentration in the seated breathing area was extremely low. Therefore, we used the CO_2_ concentration in the standing breathing area as the contaminant source concentration (*C*p) in Equation (2). Figure 7 portrays the *CRE* of the upper and lower air outlets. The *CRE* of the upper part was <1%. Notably, the *CRE* of the lower part was 1.89% in the CV car, whereas it ranged from 5.85% to 6.93% for the three ventilation rate scenarios in the SV car. The higher CRE for SV versus that of CV confirmed the advantages of SV in terms of contaminant dilution and purification. When the concentration of contaminants increases due to an increase in passengers in the car, the *CRE* will further improve.

Figure 8 portrays the temperature fields of subway cars for the four ventilation scenarios shown in Figure 5. A temperature boundary line (l_heat_) was drawn under AC and ventilation conditions in the car. The temperature of the uncooled airflow area on the left of the boundary line was mainly 300 K, whereas that on the right was 298 K, mainly representing fresh, cooled airflow. In the CV car, the temperature boundary line reached the seated passenger, indicating that the AC ventilation in the car provided poor air supply for seated passengers. In the SV car, the middle air supply airflow extended the fresh cooled air area around the seated passengers, and the temperature boundary line was far from them. Even at a ventilation rate of 7200 m^3^/h, the uncooled airflow area above 300 K in the car equipped with SV was less than that equipped with CV at the ventilation rate of 8500 m^3^/h. These findings indicate that SV could dilute the hot air area in the car more effectively due to the dispersed AC airflow. Figure 9 portrays the temperature field at the monitoring midline of the standing and seated passengers shown in Figure 8 (Figure 2b portrays the position of the monitoring midline). Figure 8 also indicates that, at heights of less than 1.4 m, the temperature of the CV car was higher than that of the SV car; at heights of more than 1.4 m, the CV created a better cooling effect due to the larger flow rate of air.

The above findings indicate that the contaminant dilution effect was slightly lower for the standing passengers in the SV car than for those in the CV car; its effect was greater for the seated passengers, and it also optimized the temperature conditions in the subway car.

### 3.2. Energy Conservation Performance

Based on the analysis in Section 2.1, we deduced that the contaminant dilution and cooling performance were better in the SV equipped with AC and the air supply flow rate of 7200 m^3^/h. We compared the summer cooling loads of the CV and SV cars for air supplies of 8500 and 7200 m^3^/h, respectively, and quantified the energy-saving effects of SV. The cooling load for a single car was itemized on an hourly basis, according to the General Code for Building Energy Conservation and Renewable Energy Utilization GB55015-2021. The calculation parameters for the interior and exterior of the cars were considered according to the Class I comfort conditions for areas that experience hot summers and cold winters, as published in the Design Code for Heating Ventilation and Air Conditioning of Civil Buildings GB50736-2012. Table 2, Table 3 and Table 4 portray the design parameters conspired in this study. As subways predominantly operate underground without solar radiation, in this study, we considered only subway cars’ unsteady heat transfer cooling load from ventilation and human heat dissipation as the important factors. Note that the heat dissipation from lights was low enough to be disregarded.

The main calculation process was as follows:

The subway car’s unsteady heat transfer cooling load, *CL*_c_, was calculated using the following equation:(3)$$CL_{c}=A\cdot \lambda \cdot \left(T_{w}-T_{n}\right)$$
where *A* is the heat transfer area of the car expressed in m^2^; a single Type-B subway car is 19 m long, 2.55 m wide, and 2.1 m high, providing a result of 101.7 m^3^. *λ* is the overall thermal transmittance of the cars’ wall (2.3 W/m^2^ K) [43]; *T*_w_ is the calculated temperature of the hourly cooling load outside the car obtained from the Design Code for Heating Ventilation and Air Conditioning in Civil Buildings GB50736-2012 and expressed in °C. T_n_ is the calculated temperature inside the car expressed in °C and obtained from CFD calculation, and subscripts 1 and 2 represent the CV and SV cars, respectively (In the following sentences, subscripts 1 and 2 have a similar meaning as just reported).

The hourly cooling load results shown in Figure 10 indicate that at 15:00 h, the maximum cooling loads of the CV and SV cars were 1.26 kW and 1.61 kW, respectively, portraying a lower average temperature in the SV car than in the CV car. In the SV car, the energy consumption for cooling, caused by the thermal transmittance from the enclosed structure, was greater than that of the CV car, primarily because the airflow within the SV car was distributed more uniformly.

#### 3.2.1. Summer Air-Conditioning (AC) Fresh-Air Cooling Load (*CL*_f_)

According to Equations (4) and (5), the fresh-air cooling load of the SV car was lower than that of the CV car, mainly because the fresh-air requirement of the SV car was lower, which reflected energy-saving characteristics of a high-temperature difference and low flow rate.
(4)$$CL_{f,1}=M_{1}\left(h_{w}-h_{n,1}\right)=\frac{2600\times 1.293}{3600}\times (97.85-52.91)=41.97\;kW$$
(5)$$CL_{f,2}=M_{2}\left(h_{w}-h_{n,2}\right)=\frac{2200\times 1.293}{3600}\times (97.85-52.91)=35.51\;kW$$
where *M* is the fresh air requirement (kg/s); *h*_w_ is the enthalpy for the calculated AC external temperature in summer, expressed as kJ/kg; and *h*_n_ is the enthalpy for the calculated parameters of the subway car in summer, expressed in kJ/kg.

#### 3.2.2. Human Heat Dissipation Cooling Load (*CL*_p_)

The unsteady heat transfer cooling and fresh-air cooling loads calculated in the previous sections are based on the calculation results of two passengers in the car, one seated and one standing, using the CFD method. Therefore, only two passengers were considered for human body heat dissipation. As the heat dissipation of the human body differs depending on the background temperature of the subway car, the heat dissipation and cooling load of the human body would also differ slightly. Notably, the temperature in the SV model was low, and the temperature of the passenger’s body surface differed from the ambient temperature of the car; therefore, the heat dissipation and cooling load were slightly higher, calculated as shown in Equations (6) and (7):(6)$$CL_{p,1}=q_{s,1}\cdot n\cdot \varphi \cdot cl_{h}=50.5\times 2\times 1\times 1=0.1\;kW$$
(7)$$CL_{p,2}=q_{s,2}\cdot n\cdot \varphi \cdot cl_{h}=60.5\times 2\times 1\times 1=0.12\;kW$$
where *q*_s_ is the normal heat dissipation of an active adult man in different room temperatures and occupations, expressed as W; when sitting still in the CV car at 301.5 K, *q*_s,1_ = 50.5 W; when sitting still in the SV car at 300 K, *q*_s,2_ = 60.5 W. n is the number of people in the subway car, n = 2; ψ is the clustering coefficient (ψ = 1). *_c_cl_h_* is the human sensible heat dissipation and cooling load coefficient, considered to be 1 in crowded places.

#### 3.2.3. Total Cooling Load (*CL*_tot_)

The cooling load of the CV car was calculated using Equation (8):(8)$$CL_{\mathrm{tot},1}=CL_{c,1}+CL_{f,1}+CL_{p,1}=1.26+41.97+0.1=43.33\;kW$$

The cooling load of the SV car was calculated using Equation (9):(9)$$CL_{\mathrm{tot},2}=CL_{c,2}+CL_{f,2}+CL_{p,2}=1.61+35.51+0.12=37.24\;kW$$
where *CL*_tot,1_, *CL*_c,1_, *CL*_f,1_, and *CL*_p,1_ represent the total cooling load, unsteady heat transfer cooling load, summer AC fresh-air cooling load, and human heat dissipation cooling load of a CV subway car, respectively. *CL*_tot,2_, *CL*_c,2_, *CL*_f,2_ and *CL*_p,2_, respectively, represent the above loads in an SV subway car.

Comparing the cooling loads for the CV and SV cars, the energy saving rate (*η*) of the SV car was 14.05%, as shown in Equation (10):(10)$$\eta =\frac{CL_{\mathrm{tot},1}-CL_{\mathrm{tot},2}}{CL_{\mathrm{tot},1}}=\frac{43.33-37.24}{43.33}\times 100\%=14.05\%$$

The cooling load composition indicates that the more uniform airflow in the SV model can reduce the ventilation flow rate and fresh air requirement of a subway car while maintaining a lower in-car temperature, causing the thermal transmittance to the outside environment through the enclosed structure to be high and increasing the heat dissipation of passengers in subway cars. This mode of low-flow and high-temperature differences observed in the SV car was helpful in reducing the fresh-air load, with its overall cooling load being lower than that of the CV car.

## 4. Discussion

Based on the different pollutant heights of standing and seated passengers, the SV technology of subway carriages was designed with top air supply and middle air supply vents for fresh air supply to standing and seated passengers, respectively. Numerical simulation showed that the CRE in the breathing zone of standing passengers was close to that of CV, and the CRE in the breathing zone of seated passengers was three times higher than that of CV during SV. In cooling mode, since the SV carriage uses a stratified supply of cold air instead of the centralized supply used in conventional carriages, the cold air supply flow was reduced by 15.3% compared to conventional carriages, with little effect on the temperature field around the passengers’ bodies. The off-peak period was modeled so that the carriages had only seated and no standing passengers. Accordingly, the upper air supply was closed, with only the central air supply required to improve the temperature field and remove pollutants, providing an energy-saving rate of 50%. Compared with the utilization of SV technology in the office, SV is more suitable for small spaces, such as subway carriages, as the pollutants are distributed at multiple fixed heights. This study suggests how SV can be used to reduce the energy consumption of rail transit air conditioning and the spread of respiratory pollutants and improve people’s confidence in choosing rail transit during the post-epidemic period. Furthermore, a good travel environment will reduce the operating cost of rail transit.

Notably, this study was based on the contaminant dilution characteristics and energy savings for one-quarter of a subway car for one seated and one standing passenger. Further studies are required to determine the effects of SV on contaminant transmission and energy conservation when more passengers are in a car while considering the movement of the passengers and the opening and closing of subway car doors.

## 5. Conclusions

The major conclusions of the study are explained below:(1)A subway car with SV could supply fresh air to seated passengers through targeted ventilation. When the ventilation rate was 7200 m^3^/h, the CRE was 5.85% for the seated passengers, which was higher than that observed in the CV (1.89%) at the ventilation rate of 8500 m^3^/h. However, the effectiveness of contaminant removal from the upper part of the car was reduced from 0.94% to 0.65%.(2)Adding a middle row of air inlets in a subway car equipped with SV caused the air supply in the car to be more uniform. The temperature field of the entire car was significantly reduced in the cooling season, and the distribution area of hot air was also reduced.(3)The hourly cooling load, according to the CFD results, indicated that the cooling energy consumption was 14.05% lower in the Type-B single subway car equipped with SV than in that equipped with CV.

## Figures and Tables

**Figure 1 ijerph-19-14521-f001:**
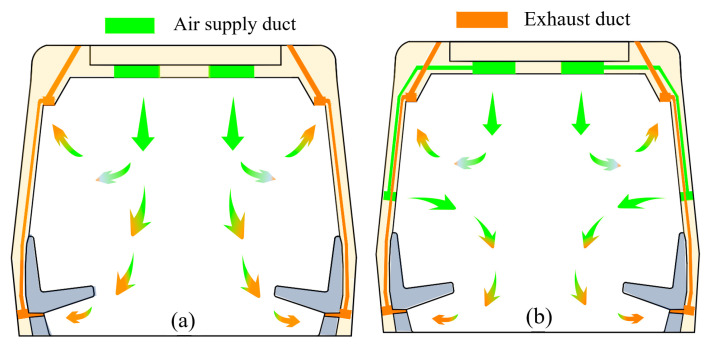
Schematic diagram of airflow organization in subway cars equipped with (**a**) conventional ventilation (CV) and (**b**) stratum ventilation (SV).

**Figure 2 ijerph-19-14521-f002:**
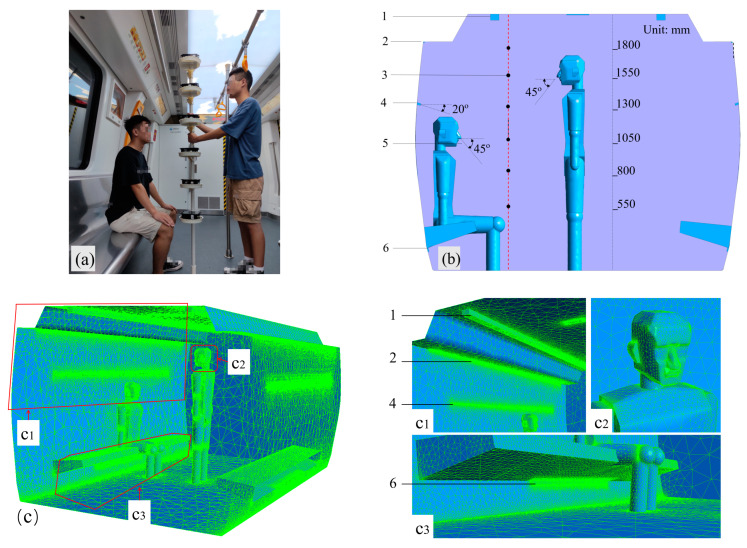
(**a**) Photograph of the field experiment and (**b**) layout of subway car ventilation system and monitoring points. (**c**) The global grid and local grid element of a carriage. (**c1**) Grid information of air exhaust outlet and air supply inlet. (**c2**) Grid information of the passengers’ head. (**c3**) Grid information of air exhaust outlet under the seat. (1) Ceiling air supply inlet, (2) air exhaust outlet at the intersection of the car body and the ceiling, (3) pollutant monitoring points and midline, (4) air supply inlet in the middle of subway body, (5) exhaled gas flow of passengers, and (6) air exhaust outlet below seats.

**Figure 3 ijerph-19-14521-f003:**
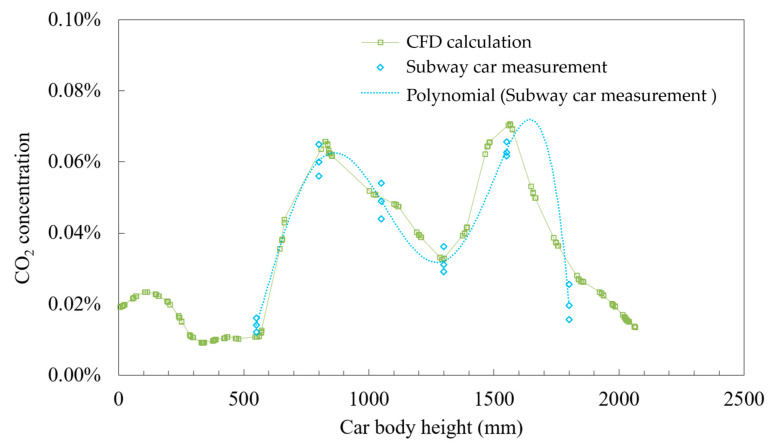
Comparison of the results of the subway car experiment and CFD test results.

**Figure 4 ijerph-19-14521-f004:**
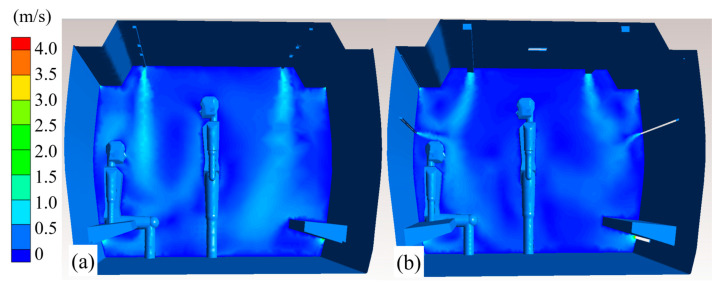
Velocity field contours of cars equipped with (**a**) CV and (**b**) SV.

**Figure 5 ijerph-19-14521-f005:**
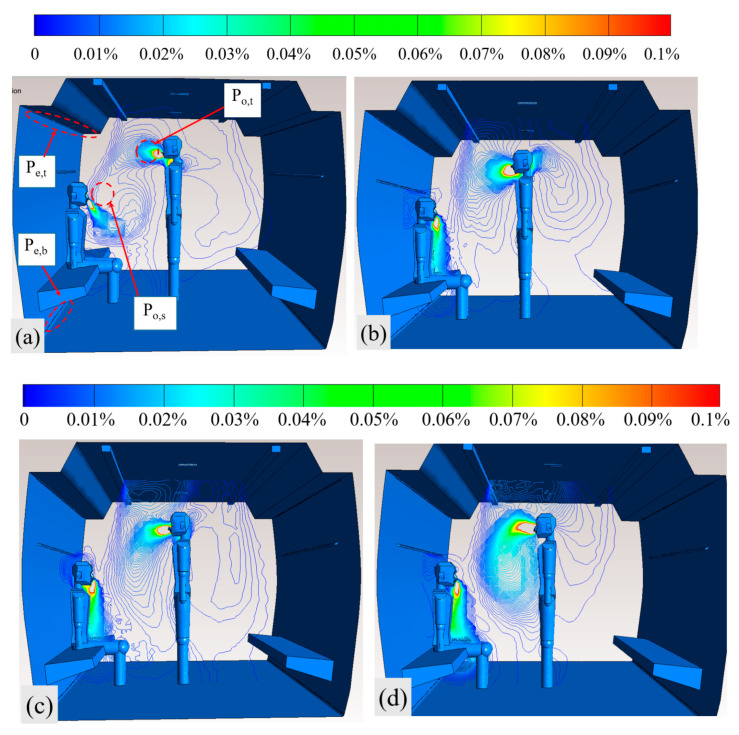
Concentrations of CO_2_ in cars equipped with (**a**) CV and (**b**–**d**) SV; *Q* = 8500, 8500, 7850, and 7200 m^3^/h, respectively (**a**–**d**).

**Figure 6 ijerph-19-14521-f006:**
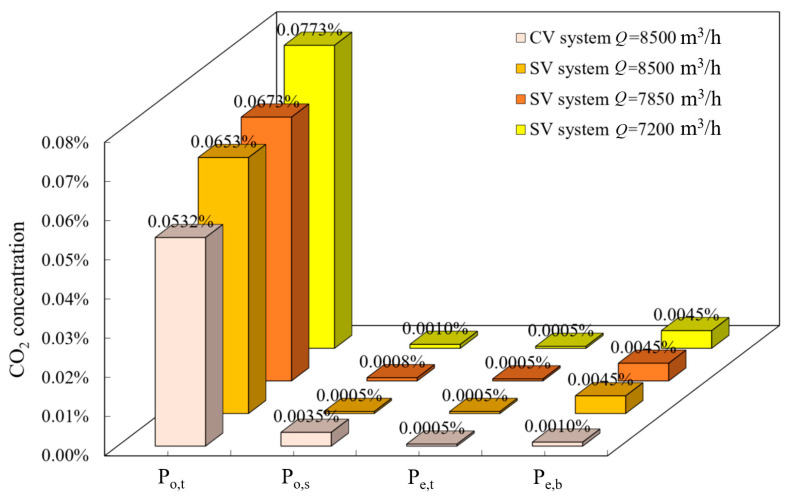
Concentrations of CO_2_ at various monitored points.

**Figure 7 ijerph-19-14521-f007:**
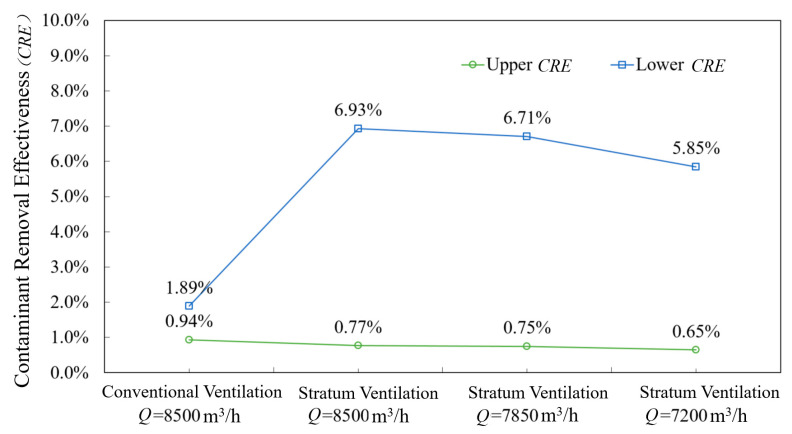
Comparison of contaminant removal between the two ventilation methods (CV and SV).

**Figure 8 ijerph-19-14521-f008:**
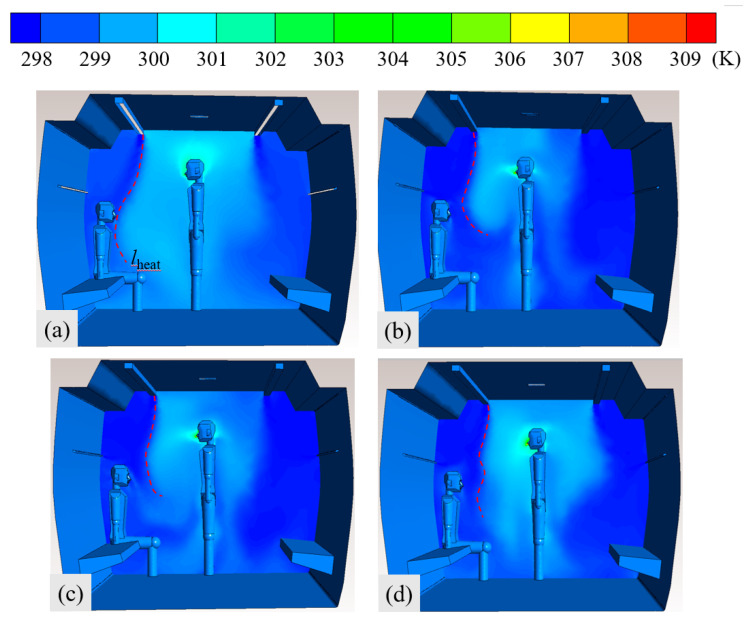
Temperature fields of cars under conventional and SV for *Q* = (**a**) 8500, (**b**) 8500, (**c**) 7850, and (**d**) 7200 m^3^/h.

**Figure 9 ijerph-19-14521-f009:**
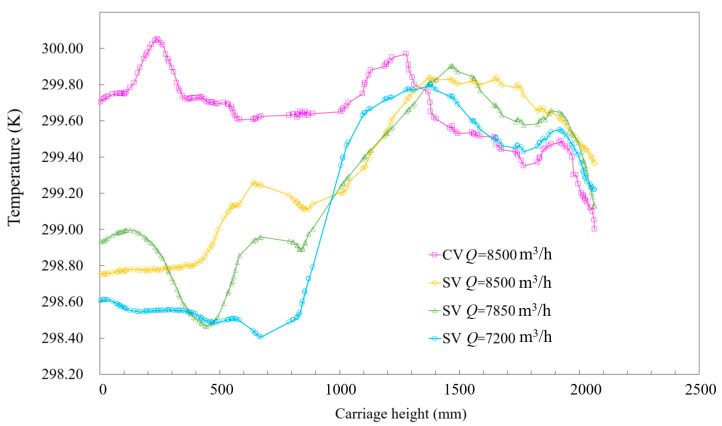
Comparison of temperatures at the midline between the seated and standing passengers for different *Q* values.

**Figure 10 ijerph-19-14521-f010:**
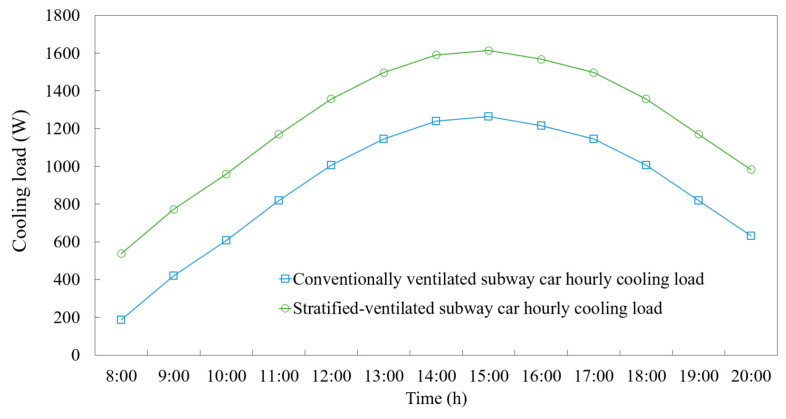
Hourly cooling loads of the subway cars equipped with CV and SV under representative summer weather conditions.

**Table 1 ijerph-19-14521-t001:** Boundary and initial conditions for CFD calculations.

Boundary Conditions	Type	Parameters	Initial Conditions	Parameters
Ceiling air supply inlet	Mass flow inlet	0.76 kg/s, 298 K	Internal CO_2_	0%
Air exhaust outlet at the intersection of the car body and ceiling	Pressure outlet	0 Pa, 298 K	Internal N_2_	78%
Air supply inlet in the middle of subway body	Mass flow inlet	0.76 kg/s, 298 K	Internal O_2_	22%
Air exhaust outlet below seats	Pressure outlet	0 Pa, 298 K	Passenger exhaled CO_2_	4%
Exhaled gas flow of passengers	Mass flow inlet	1.07 × 10^−4^ kg/s, 309 K	Internal initial temperature	309 K

**Table 2 ijerph-19-14521-t002:** Calculation parameters of air in the cars equipped with SV and CV in summer.

Parameter	CV	SV
Temperature of air supply (K)	298	298
Average in-car temperature (K)	301.5	300
Design temperature (K)	299	299
Relative humidity (%)	50	50
Air enthalpy (kJ·kg^−1^)	52.91	52.91
Ventilation flow rate (m^3^·h^−1^)	8500	7200
Fresh air requirement (m^3^·h^−1^)	2600	2200

**Table 3 ijerph-19-14521-t003:** Calculation parameters of air outside the subway car in summer.

Temperature (K)	Relative Humidity (%)	Air Density (kg·m^−3^)	Enthalpy (kJ·kg^−1^)
308	68	1.293	97.85

**Table 4 ijerph-19-14521-t004:** Hourly temperature inside and outside the subway carriage under representative summer weather conditions.

Time (h)	8:00	9:00	10:00	11:00	12:00	13:00	14:00	15:00	16:00	17:00	18:00	19:00	20:00
Tw/K	302.3	303.3	304.1	305	305.8	306.4	306.8	306.9	306.7	306.4	305.8	305	304.2
Tn,1/K	301.5	301.5	301.5	301.5	301.5	301.5	301.5	301.5	301.5	301.5	301.5	301.5	301.5
Tn,2/K	300	300	300	300	300	300	300	300	300	300	300	300	300

## Data Availability

The data presented in this study are available on request from the corresponding authors.

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
