# Peer review of "Stratum Ventilation: Enabling Simultaneous Energy Conservation and Air Purification in Subway Cars"

_ijerph, 2022, doi:10.3390/ijerph192114521_

Round 1
Reviewer 1 Report
In this paper, a ventilation scheme was simulated using CFD for efficient ventilation of the ventilation system of a subway vehicle.
The summary of the paper is as follows.
In the introduction part, it is appropriate to introduce the ventilation model in vehicles and offices, which has been studied previously, and to explain the differences from this study.
In the research method, CFD simulation method and boundary condition were introduced in relatively detail.
In the model verification, the error rate of the model was confirmed by checking how much the difference between the simulation and the actual value is shown by using sensors.
In the results, the pollutant removal effect (CRE) was confirmed by simulation using various methods, and the optimal breathing conditions were confirmed through four ventilation scenarios.
Also, by checking the energy saving method, the cooling load and external air enrichment were confirmed.
Two results (dilution purification performance, energy saving performance) were presented at once however, a comprehensive discussion section about two results together is needed to clear overall results.
In conclusion, the results are well expressed in abbreviated form.
The reference part needs to be corrected by rechecking the MDPI form.
Overall, it can be said that this paper is sufficient to be published in the atmosphere journal if a comprehensive discussion section is added.
Reviewer 2 Report
The authors proposed a stratum ventilation technology for subway cars. Computational fluid dynamics was used to compare the energy conservation and air purification performance between traditional subway cars and those with stratum ventilation. This study embodied the new application of stratum ventilation technology. A minor revision is recommended for the manuscript.
1. The literature review in the introduction did not cover the latest research on stratum ventilation, which was not conducive to reflecting the value of this study.
2. To clearly show the working principle of air conditioning in subway cars, the indicative colors of the air conditioning supply duct and exhaust duct in Figure.1 should preferably be consistent with those of the fresh air and contaminated air in the carriage.
3. The pollutants exhaled from the human mouth include not only a gas phase but also liquid droplets. Why did this study only analyze carbon dioxide?
4. Why did the CFD simulation in this manuscript utilize a model with one person standing and the other sitting? The reasons need to be further clarified.
5. The image of mesh quality information should be added to the manuscript.
Reviewer 3 Report
This paper compared the velocity fields of conventional ventilation and stratum ventilation in subway cars to understand the differences in their airflow organizations and contaminant removal efficiencies, along with the energy savings of four ventilation scenarios. The paper is well-structured. However, a minor revision is needed before it is considered to be accepted:
(1) Some abbreviations in the text need only be marked the first time they appear and do not need to be repeated, such as SV,CV.
(2) Units of time need to be written in Figure 10 and Table 4?
(3) If there are many people in the car during rush hour, whether the air flow and test data will have a great impact?
(4) Please check carefully and revise the format of the reference.
(5) Authors can cite some relevant reports to demonstrate the threat of air pollution to human health and ecosystems (e.g. Journal of Materials Chemistry A, 2020, 8(36): 18955-18962.; Small, 2017, 13(46): 1702139.; Journal of Cleaner Production, 2021, 284: 124761.; Sustainable Cities and Society, 2019, 49: 101569; Energy and Buildings, 2017, 151: 217-227.) .
